# Nutritional status and treatment outcomes of tuberculosis in Mizan Tepi University Teaching Hospital, a five -year retrospective study

**Asnake Simieneh**[1]*, **Surafel Gashaneh**[2], **Rahel Dereje**[3]

**1** Department of Medical Laboratory Sciences, College of Medicine and Health Sciences, Wolkite University, Wolkite, Ethiopia, **2** Department of Medical Laboratory Sciences, College of Medicine and Health Sciences, Mizan Tepi University, Mizan-Aman, Ethiopia, **3** Department of Public Health, School of Public Health, College of Medicine and Health Sciences, Mizan Tepi University, Mizan-Aman, Ethiopia

* asnakesimieneh@gmail.com

**Data Availability Statement:** All relevant data are within the manuscript and its Supporting Information files.

## Abstract

### Background

Public health problems related to tuberculosis (TB) remain substantial globally, particularly in resource-limited countries. Determining TB treatment outcomes and identifying contributing factors are the basic components of the TB control strategy. In Ethiopia, different studies have been done on treatment outcomes and multiple associated factors, and there is also a little information on the effect of nutritional status on TB treatment outcomes. So there is a need for comprehensive research that examines the combined effects of multiple factors along with nutritional status.

### Methods

A five-year institution-based retrospective cross-sectional study was conducted at Mizan Tepi University Teaching Hospital, South West Ethiopia. This study included all tuberculosis patients who were documented in the TB registration and had known treatment outcomes at the treatment facility between January 1, 2016, and December 31, 2020. Data was collected through a pretested structured data extraction checklist. Data were entered into Epidata version 3.1 and analyzed through SPSS version 22. Multiple logistic regression was employed to assess the association between dependent and independent variables. A p-value of less than 0.05 was considered statistically significant.

### Result

Of the total 625 TB patients, 283 (45.3%), 175 (28%), and 167 (26.7%) had smear-positive, extra-pulmonary, and smear-negative tuberculosis, respectively. The majority of study participants had normal weight (62.2%), were in the age group of 15–44 (67.4%), were new cases (73.8%), and were from urban areas (69.4%). About 32.2% of cases were HIV-positive. The overall unsuccessful treatment rate was 25%. From the total unsuccessful treatment rates, the highest proportion was a death rate of 90 (14.4%), followed by a treatment failure of 56 (9%). Being female (AOR = 1.7, 95% CI: 1.2–2.5), HIV positive (AOR = 2.7, 95% CI: 1.9–4.1), undernutrition (BMI<18.5kg/m$^2$) (AOR = 1.9, 95% CI: 1.3–2.9), and

**Funding:** The author(s) received no specific funding for this work.

**Competing interests:** The authors have declared that no competing interests exist.

smear-negative pulmonary TB (AOR = 1.6, 95% CI: 1–2.5) were independent predictors of unsuccessful treatment outcomes.

## Conclusion

The treatment success rate in the study area is very poor. Poor treatment outcomes were associated with undernutrition, female gender, HIV positivity and smear-negative pulmonary TB. So, continuous and serious supervision and monitoring of directly observed treatment short course (DOTS) program accomplishment, early detection of HIV and TB, prompt anti TB and antiretroviral treatment initiation and adherence, enhanced nutritional assessment, and counseling services need to be strengthened to improve treatment outcomes.

## Introduction

Tuberculosis (TB) is a communicable disease that is a major cause of ill health and one of the leading causes of death worldwide. Tuberculosis is the 2nd leading cause of death from a single infectious agent, next to coronavirus (COVID-19) and above HIV/AIDS [1]. Even though highly effective treatments have been available for decades, tuberculosis continues to be a challenge, especially in low-income countries such as African countries [2].

According to a World Health Organization (WHO) report, in 2021, an estimated 10.6 million people globally became infected with tuberculosis, with a mortality rate of around 1.6 million people, including those who are living with HIV (187,000). The mortality and morbidity due to tuberculosis are high in 2021 compared to a report in 2020. In 2020, there were an estimated 10.1 million and 1.5 million cases of TB morbidity and mortality worldwide, respectively [1]. In Ethiopia, the leading cause of death from a single infectious agent is tuberculosis [3]. It is also among the 30 countries with the highest triple burden of tuberculosis (TB), TB-HIV co-infection, and multidrug-resistant TB (MDR-TB) globally. Ethiopia is also known for a high proportion of extrapulmonary TB (EPTB), which accounts for more than 33% of all forms of TB [4].

According to Ethiopian national guidelines for the management of TB, the foundational work in TB infection control is early and rapid diagnosis and effective treatment of TB patients. Regular monitoring and evaluation of tuberculosis treatment outcomes helps assess TB disease control programs. This continuous monitoring and evaluation program is a key component of a TB program called the directly observed treatment short course (DOTS). Ethiopia has adopted the DOTS strategy since 1997 and plans to successfully treat 90% of confirmed TB cases and enroll them in the DOTS program [5]. According to the 2021 WHO report, the successful TB treatment outcome rate in Ethiopia was 86% by 2020 [1]. Research conducted in Ethiopia has revealed varying percentages of successful tuberculosis treatment, ranging from 74.2% to 93.8% [6–10].

Unfavorable TB treatment outcomes are associated with many risk factors. A study done in Uzbekistan showed that being above 55 years of age, HIV-positive, sputum smear-positive, previously treated, and being jobless were independent predictors of unfavorable TB treatment outcomes [11]. Another study in Ethiopia showed that there is a significant association between residing in rural areas, being smear-negative PTB and EPTB, and unsuccessful TB treatment outcomes [10].

The immune system is weakened by undernutrition, which raises the possibility that latent TB may become active TB. Undernutrition may also result in unsuccessful treatment outcomes

through delayed sputum conversion during treatment [12]. Different studies in different parts of the world indicated a significant association between undernutrition and unsuccessful TB treatment outcomes [12–14]. A study in India reported that undernutrition before the onset of TB disease and during anti-TB treatment initiation doubled the risk of unfavorable outcomes [13].

There are studies done on the association between unsuccessful TB treatment outcomes and associated factors in different parts of Ethiopia, such as Sekota [8], Dilla [10], Wolayta [15], Jimma [6, 7], Gambella [9], and Madda Walabu [16]. There is also little information on the effect of nutritional status on TB treatment outcomes [17]. However, to the knowledge of the author, there is no study that examines the combined effect of multiple factors along with nutritional status in Ethiopia. So there is a need for comprehensive research that examines the combined effects of multiple factors along with nutritional status. Furthermore, for more effective tuberculosis disease management and control initiatives, it is crucial to determine TB trends and treatment outcomes in healthcare facilities. So, this study aimed to assess the effect of nutritional status along with other multiple factors on TB treatment outcomes.

## Methods

### Study area and period

The study was conducted at the Mizan-Tepi University Teaching Hospital (MTUTH) TB clinic found in Mizan Aman town, which is 565 km away from Addis Ababa. The hospital has antenatal care (ANC), outpatient, inpatient (medical and surgical), emergency, psychiatry, a TB clinic, and youth-friendly or counseling wards. The TB clinic started working in 2012 with the registration of necessary information in the logbook. The data was collected from August 10th to 30th, 2021, on tuberculosis patients who had registered treatment outcomes between January 1st, 2016 and December 31st, 2020.

### Study design

Institution-based retrospective cross-sectional study design was conducted.

### Population

**Study population.** The study participants were all patients who registered for anti-TB treatment with full recorded information from January 1st / 2016 to December 31st / 2020.

### Inclusion and exclusion criteria

All TB patients who had complete information from January 1st / 2016 to December 31st / 2020 were included in the study and those patients who have missed information and transferred out cases were excluded.

### Sample size and sampling technique

A total of 676 tuberculosis patients were treated from January 1st / 2016 to December 31st / 2020 at Mizan Tepi University Teaching Hospital. Of these, only 625TB patients fulfilled the inclusion criteria and were included in the study (Fig 1).

### Study variables

The dependent variable was TB treatment outcomes and the independent variables were age, gender, residence, HIV status, type of TB, category of TB, and nutritional status.

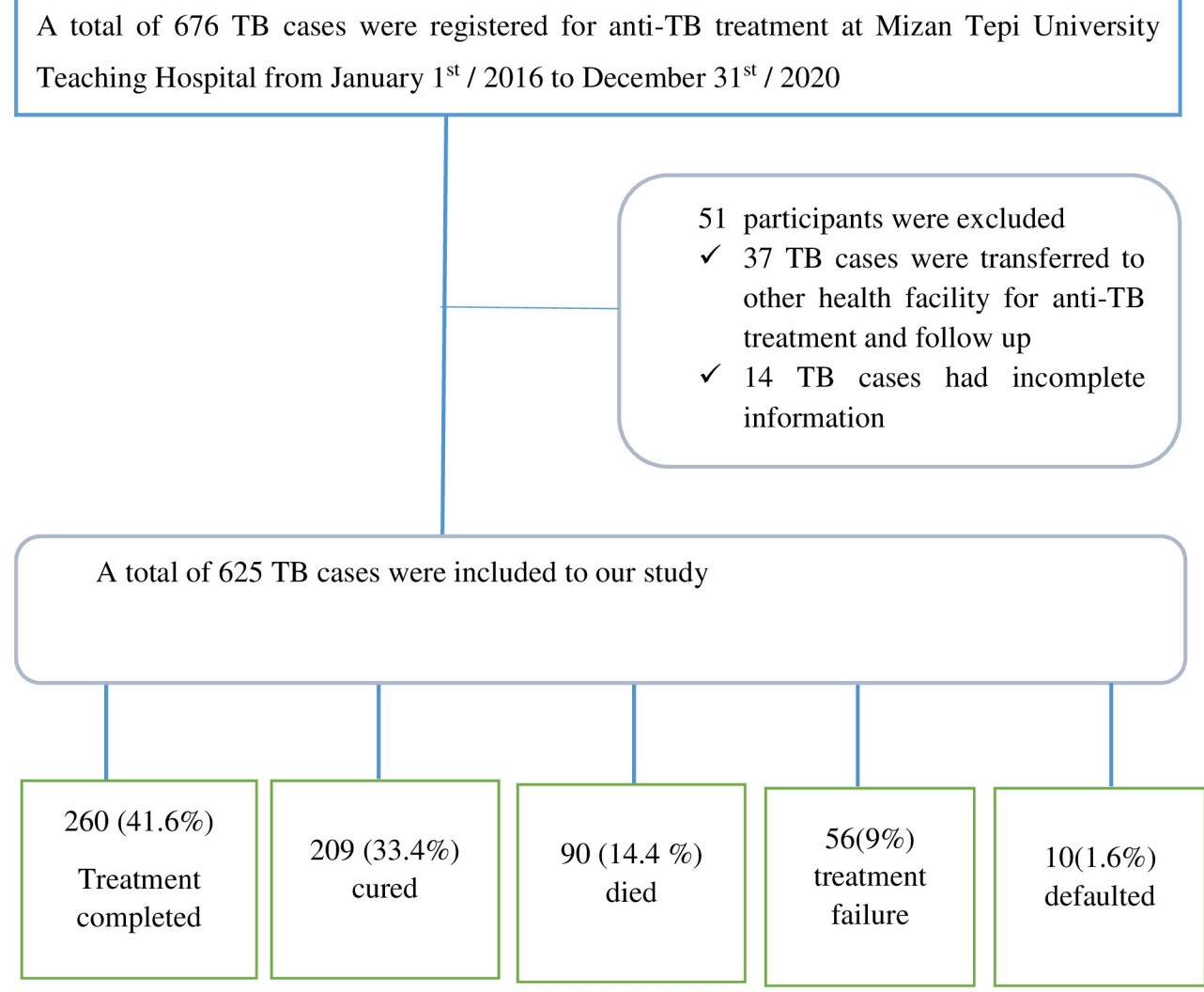

**Fig 1. Flow chart showing overall participant screening process and treatment outcomes classification result.**

## Data collection period, procedures, and data quality

Data was collected from the TB register of patients using a prepared standard checklist by the data collectors **from August 10th to 30th/2021 for research purposes.** The contents of the checklist included socio-demographic characteristics, HIV status, types and categories of TB, nutritional status, and treatment outcome (S2 File). Nutritional status was determined by the body mass index (BMI) that was measured during the initiation of anti-TB treatment. To assure the quality of the data, it was collected by two laboratory professionals after one-day training on the data collection format and techniques of data collection. The data was checked for its completeness every day by the principal investigator.

## Data processing and analysis

The data were entered and cleaned by Epidata version 3.1, and then exported to SPSS version 22. Descriptive analyses such as frequencies, percentages, means, and SD were done. Undernutrition was defined as a BMI<18.5 kg/m$^2$.

Bivariate and multivariable logistic regression were done to see the association between the outcome variable and independent variables. During bivariate analysis, variables that had a p-value less than 0.25 were selected and entered into the multivariable logistic regression. The Hosmer -Lemeshow of the goodness of fit test was checked to verify the model fitness (p-value = 0.37), multicollinearity was assessed, and all independent variables were not collinear (VIF >10). The variables with a p-value less than 0.05 were taken as significant values with an OR and 95% CI.

## Operational definition

The following definitions are based on National Tuberculosis and Leprosy Control Program (NTLCP) guidelines and World Health Organization definitions and reporting framework for tuberculosis, 2013 revision [18].

**Smear-positive pulmonary TB.** A patient with at least two sputum samples that were microscopically positive for acid-fast bacilli (AFB) or a patient with abnormal chest radiographs indicative of active pulmonary TB and one sample of sputum that was microscopically positive for AFB.

**Smear- negative pulmonary TB.** A patient with abnormal chest radiographs and symptoms suggestive of active pulmonary TB with at least two negative sputum specimens for AFB by microscopy.

**Extra-pulmonary TB (EPTB).** Tuberculosis affecting organs other than the lungs, such as the lymph nodes, abdomen, genitourinary system, skin, joints, and bones, as well as the meninges.

**New case.** New cases are those in which the patient has never received TB treatment or has only recently begun (less than 1 month) taking anti-TB medications.

**Previously treated case.** When patients have received 1 month or more of anti-TB drugs in the past, may have positive or negative bacteriology, and may have a disease at any anatomical site.

**Cure.** Cure entails a patient whose initial sputum smear or culture result was positive but who had negative results in the final month of treatment and on at least one prior occasion.

**Treatment completed.** A patient without a negative sputum smear or culture results in the last month of treatment and on at least one previous occasion after treatment completion.

**Treatment failure.** A patient whose sputum smear or culture is positive at 5 months or later during treatment.

**Died.** When a patient dies for any reason during the course of treatment.

**Default.** A patient whose treatment was interrupted for more than two months in a row.

**Transfer out.** A patient who has been transferred to another health care setting and whose treatment outcome has not been determined.

**Treatment success.** A total of cases that have had treatment completed and cured outcomes.

**Unsuccessful TB treatment.** A total of cases that have died, treatment failure, and default outcomes.

In this study, nutritional status was determined by the body mass index (BMI) [19].

**Underweight.** If the BMI measurement is below 18.5 kg/m$^2$.

**Normal weight.** If the BMI measurement is between 18.5–24.9 kg/m$^2$.

**Overweight.** If the BMI measurement is between 25–29.9 kg/m$^2$.

**Obesity.** If the BMI measurement is greater than 30 kg/m$^2$.

## Ethical consideration

The study was ethically approved by the Institutional Review Board of Mizan Tepi University College of Medicine and Health Sciences (IRB/379/2021). Permission was obtained from the

hospital administration before data collection. To maintain confidentiality, the data was anonymized and de-identified before being analyzed.

## Result

### Socio-demographic characteristics of study participants

Overall, 625 documents were reviewed. Among the study participants, 317 (50.7%) were males, and the rest, 308 (49.3%), were females. More than two-thirds of participants were in the age group between 15 and 44. The majority of the study participants were from urban areas (69.4%).

As "Table 1" shows, about 389 (62.2%) and 191 (30.6%) participants were of normal weight and underweight, respectively. About 32 (5.1%) of the participants were overweight, while 13 (2.1%) were obese. Among the tuberculosis patients, 450 (72%) had pulmonary tuberculosis, and 175 (38%) had extra-pulmonary tuberculosis. Among the total TB cases, 283 (45.3%) were smear-positive pulmonary TB.

### Treatment outcome of participants

Our study showed that the majority of the participants, 260 (41.6%), completed the treatment, and 209 (33.4%) were cured. Among the participants, 90 (14.4%) of the participants have died, 56(9%) of the participants had treatment failure. About 10 (1.6%) of the participants had defaulted on the treatment. Overall, 469 (75%) participants had successful treatment outcomes, and the rest, 156 (25%), had unsuccessful treatment outcomes (Table 2).

### Factors associated with unsuccessful treatment outcome of tuberculosis

All variables were entered into binary logistic regression, and variables with a p-value <0.25 were selected. Variables including sex, residence, type of TB, HIV status, category of TB, and

**Table 1. Socio-demographic and clinical characteristics of registered TB patients in Mizan Tepi University Teaching Hospital, southwest Ethiopia 2016–2020.**

| Variables | | Frequency | Percentage (%) |
|---|---|---|---|
| Sex | Female | 308 | 49.3 |
| | Male | 317 | 50.7 |
| Age | <15 | 67 | 10.7 |
| | 15–44 | 421 | 67.4 |
| | 45–64 | 109 | 17.4 |
| | ≥65 | 28 | 4.5 |
| Residence | Urban | 434 | 69.4 |
| | Rural | 191 | 30.6 |
| BMI | Underweight | 191 | 30.6 |
| | Normal weight | 389 | 62.2 |
| | Overweight | 32 | 5.1 |
| | Obese | 13 | 2.1 |
| Type of TB | Smear Positive PTB | 283 | 45.3 |
| | Smear Negative PTB | 167 | 26.7 |
| | EPTB | 175 | 28.0 |
| HIV status | Positive | 201 | 32.2 |
| | Negative | 424 | 67.8 |
| Category of TB | New | 461 | 73.8 |
| | Previously treated | 164 | 26.2 |

BMI; body mass index, EPTB; extra pulmonary tuberculosis, PTB; pulmonary tuberculosis

**Table 2. Treatment outcome of TB cases in Mizan Tepi University Teaching Hospital, Southwest Ethiopia, from 2016–2020.**

| Variable | | Frequency(n = 625) | Percentage (100%) |
|---|---|---|---|
| Treatment outcome | Treatment completed | 260 | 41.6 |
| | Cured | 209 | 33.4 |
| | Death | 90 | 14.4 |
| | Treatment failure | 56 | 9 |
| | Default | 10 | 1.6 |
| The success of treatment outcome | Successful | 469 | 75 |
| | Unsuccessful | 156 | 25 |

BMI were found to be significantly positively associated with unsuccessful TB treatment outcomes. Then multivariable logistic regression was conducted, and variables including female sex, being HIV positive, being smear-negative PTB, being underweight, and being previously treated were independent predictors of unsuccessful TB treatment outcomes (Table 3).

## Discussion

In this research, data for 625 participants was collected from the TB registration record at the DOT clinic of Mizan Tepi University Teaching Hospital. Males were nearly equal to females in this study, accounting for 50.7%. In contrast to this study, previous studies reported that males were dominant, such as a study in Dilla [10] and Jimma [6, 7]. The age range that is most economically productive, 15 to 44 years old, is the one where tuberculosis prevalence was highest in our study. Our finding was in agreement with previous studies in Denmark [20], Wolayta [15], and Jimma [6]. This may show the negative effect of TB on the social and economic development of society.

**Table 3. Bivariate and multivariable logistic regression analysis of unsuccessful treatment outcome of tuberculosis and associated factors in Mizan Tepi University Teaching Hospital, Southwest Ethiopia, from 2016–2020.**

| Variables | | Treatment outcome | | COR(95% CI) | AOR(95%CI) | P-value |
|---|---|---|---|---|---|---|
| | | Successful | Unsuccessful | | | |
| Sex | Male | 255 | 62 | 1 | 1 | |
| | Female | 214 | 94 | 1.8(1.2–2.6) | 1.7(1.2–2.5) | **0.005** |
| Residence | Urban | 332 | 102 | 1 | 1 | |
| | Rural | 137 | 54 | 1.2(0.8–1.8) | 1.2(0.8–1.9) | **0.26** |
| Type of TB | Smear Positive PTB | 221 | 62 | 1 | 1 | |
| | Smear negative PTB | 115 | 52 | 1.6(1–2.4) | 1.6(1–2.5) | **0.042** |
| | EPTB | 113 | 42 | 1.1(0.7–1.7) | 1.1(0.68–1.75) | **0.706** |
| HIV status | Negative | 347 | 77 | 1 | 1 | |
| | Positive | 122 | 79 | 2.9 (2–4.2) | 2.7(1.9–4.1) | **0.001** |
| Category of TB | New | 358 | 103 | 1 | 1 | |
| | Previously treated | 111 | 53 | 1.6 (1.1–2.4) | 1.7(1.2–2.6) | **0.013** |
| BMI | Normal weight (BMI,18.5–24.9Kg/m$^2$) | 306 | 83 | 1 | 1 | |
| | Underweight BMI<18.5 | 129 | 62 | 1.7(1.2–2.6) | 1.9 (1.3–2.9) | **0.02** |
| | Overweight BMI,25–29.9 | 25 | 7 | 1.0(0.4–2.4) | 1.2(0.5–3.0) | **0.649** |
| | Obese BMI>30 | 9 | 4 | 1.6 (0.5–3.4) | 1.7(0.49–4.6) | **0.392** |

BMI; body mass index, EPTB; extra pulmonary tuberculosis, PTB; pulmonary tuberculosis

In our study, the magnitude of treatment completion and cure were 41.6% and 33.4%, respectively, with an overall treatment success rate of 75%. The treatment success rate obtained in this study was very much lower than a study done in different parts of Ethiopia, such as 93.8% in Sekota [8], 92.5% in Harar [21], 91.9% in East Wollega [22], 90.1% in Debre Tabor [23], 89.7% in Southwest Ethiopia [24], and 85.2% in Dilla [10]. The treatment success rate in our study was also lower compared to the WHO treatment success report for Ethiopia, 86% [1], and the WHO 2030 international target of 90% [5]. The low treatment success rate in this study compared to the other studies might be due to poor implementation of the DOTs strategy, high HIV co-infection (32.2%), delays in treatment initiation, and the high death rate in our study, which accounts for about 14.4%.

In this study, the magnitude of death, treatment failure, and default were 14.4%, 9%, and 1.6%, respectively, with an overall unsuccessful treatment rate of 25%. The result derived from this study is higher than the unsuccessful treatment rate reported from Woldia (19.3%) [25], Dilla (14.8%) [10], Hawassa (7.6%) [26], Sekota (6.2) [8] and Pakistan (5.1%) [27]. The death rate in this study comprised the majority of the unsuccessful rate of TB treatment. The high death rate may be due to treatment failure, high HIV co-infection, and delays in anti-TB treatment initiation. A study in southwest Ethiopia showed an association between delays in anti-TB treatment initiation more than 30 days after the onset of infection and the result of death [24]. Another study in southern Ethiopia showed that there is a high risk of death and unsuccessful TB treatment outcomes due to TB-HIV co-infection [28]. The treatment failure rate comprised the second-largest portion of the unsuccessful rate in TB treatment. The treatment failure rate in this study is higher than a report from Debre Tabor (3.5%) [29], Tigray Region (3.7%) [30], and Dilla (0.3%) [10]. The high rate of treatment failure in our study may be attributable to poor counseling and adherence to anti-TB treatments in the community.

According to our study findings, female TB patients have a higher chance of unsuccessful treatment than male TB patients. Studies conducted at Woldia [25] and Afar [31] supported this finding. The unsuccessful treatment rate in females might be due to different maternal-related complications and less immunity in females compared to males. Moreover, a study in Egypt [32] reported that there is a higher risk of anti-TB drug complications among females than males.

Our study showed that smear-negative pulmonary TB patients had a significantly higher unsuccessful treatment rate compared to smear-positive pulmonary TB patients. This finding is consistent with previous studies in Jimma [6], Dilla [10], and Eastern Ethiopia [33]. This may be due to the high rate of HIV-TB co-infection among smear-negative and extra-pulmonary TB patients, which may increase unsuccessful treatment rates such as death and treatment failure. A study in eastern Ethiopia explains that there is a probability of misdiagnosis in smear-negative patients, which resultes in poor treatment response [33].

In this study, TB cases that are underweight were nearly twice as likely as those of normal weight to have unsuccessful treatment outcomes. This finding is consistent with studies done in Addis Ababa [17] and India [13]. This could be due to the reduced absorption of rifampin and isoniazid or increased toxicity from drugs like ethambutol and aminoglycosides in undernourished TB patients.

Our study shows that HIV-positive TB patients have a 2.7 times higher risk of developing unsuccessful treatment rates than HIV-negative TB patients. A consistent finding was found in Malaysia [34], Portugal [35], Addis Ababa [36], Harar [21], and Eastern Ethiopia [33]. It could be due to the fact that HIV-positive individuals have suppressed immunity, and they may not take the anti-TB drugs as recommended because of concerns related to drug interactions and adverse effects. HIV continues to pose a double challenge, both increasing the risk of TB infection and unsuccessful treatment outcomes [37].

## Limitations of the study

Our study is a retrospective study based on secondary data, so it was not possible to determine all factors, and some significant variables such as comorbidity with chronic illness, distance from the treatment center, occupation, educational level, and economic status of patients were missed that could affect both the nutritional status and the patient's treatment outcomes.

## Conclusion and recommendation

The overall success rate of TB treatment in the study area was very unsatisfactory (75%), which is below the WHO end TB strategy (90%) and from the national treatment success rate of Ethiopia (86%). Death and treatment failure comprised the major portion of the unsuccessful treatment rate (14.4%) and (9%), respectively, which is a severe public health issue that requires immediate attention. Being female, having smear-negative pulmonary TB, being HIV positive, being previously treated, and being underweight were significantly associated with unsuccessful treatment outcomes.

Thus, continuous and regular supervision and monitoring of DOTS program accomplishment, early detection of HIV and TB, prompt anti-TB and antiretroviral treatment initiation and adherence, and counseling services need to be strengthened. Additionally, it is important for policymakers to focus on addressing undernutrition in TB treatment programs, incorporating nutrition support initiatives, integrating nutrition evaluation and counseling, and ensuring TB patients have access to nourishing food.

## Supporting information

**S1 File. The original SPSS dataset used and analyzed in our study.**
(SAV)

**S2 File. The standard checklist that we used to extract the data from the TB registration book.**
(DOCX)

## Acknowledgments

The authors are grateful to the study participants for participating in this research. We are also thankful to the staff of Mizan Tepi University Teaching Hospital TB Clinic for their assistance and guidance during data collection.

## Author Contributions

**Conceptualization:** Asnake Simieneh, Surafel Gashaneh, Rahel Dereje.

**Data curation:** Asnake Simieneh, Surafel Gashaneh, Rahel Dereje.

**Formal analysis:** Asnake Simieneh, Surafel Gashaneh, Rahel Dereje.

**Funding acquisition:** Rahel Dereje.

**Investigation:** Asnake Simieneh, Surafel Gashaneh, Rahel Dereje.

**Methodology:** Asnake Simieneh.

**Project administration:** Asnake Simieneh, Surafel Gashaneh.

**Resources:** Asnake Simieneh, Rahel Dereje.

**Software:** Asnake Simieneh, Rahel Dereje.

**Supervision:** Asnake Simieneh, Rahel Dereje.

**Validation:** Asnake Simieneh, Rahel Dereje.

**Visualization:** Asnake Simieneh, Surafel Gashaneh.

**Writing – original draft:** Asnake Simieneh, Surafel Gashaneh.

**Writing – review & editing:** Asnake Simieneh, Rahel Dereje.

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
