## [Decision Letter · Decision Letter 0]

23 Nov 2023

PONE-D-23-16197Nutritional Status and Treatment Outcomes of Tuberculosis in Mizan Tepi University Teaching Hospital - A five year Retrospective Study

PLOS ONE Dear Dr. Simieneh, 

Thank you for submitting your manuscript to PLOS ONE. After careful consideration, we feel that it has merit but does not fully meet PLOS ONE’s publication criteria as it currently stands. Therefore, we invite you to submit a revised version of the manuscript that addresses the points raised during the review process.

We look forward to receiving your revised manuscript.

Kind regards,

Mengistu Hailemariam Zenebe, PhD

Academic Editor

PLOS ONE

Journal Requirements:

The name of the colleague or the details of the professional service that edited your manuscript.A copy of your manuscript showing your changes by either highlighting them or using track changes (uploaded as a *supporting information* file).A clean copy of the edited manuscript (uploaded as the new *manuscript* file).

https://journals.plos.org/plosone/article?id=10.1371/journal.pone.0150560

In your revision ensure you cite all your sources (including your own works), and quote or rephrase any duplicated text outside the methods section. Further consideration is dependent on these concerns being addressed.

6. We note you have included a table to which you do not refer in the text of your manuscript. Please ensure that you refer to Table 2 in your text; if accepted, production will need this reference to link the reader to the Table.

**Additional Editor Comments:**

dear Author,

I can say just have the amendment on the given comments from the reviewers staring from the title name correction (if you accept) up to the reference writing.

hope you will let us see the update soon

Best

Reviewers' comments:

Reviewer's Responses to Questions

**Comments to the Author**

1. Is the manuscript technically sound, and do the data support the conclusions?

Reviewer #1: No

Reviewer #2: Yes

2. Has the statistical analysis been performed appropriately and rigorously? 

Reviewer #1: I Don't Know

Reviewer #2: Yes

3. Have the authors made all data underlying the findings in their manuscript fully available?

Reviewer #1: No

Reviewer #2: No

4. Is the manuscript presented in an intelligible fashion and written in standard English?

Reviewer #1: No

Reviewer #2: Yes

5. Review Comments to the Author

Reviewer #1: Research title: Nutritional Status and Treatment Outcomes of Tuberculosis in Mizan Tepi University Teaching Hospital - A five year Retrospective Study

General comment

The research title is highly relevant and scientifically significant. However, the research design and the approach to conducting the study may not be appropriate for effectively addressing the research questions. Additionally, the manuscript's writing style appears outdated and does not align with modern standards. Significant revisions are required for the manuscript. Furthermore, the English writing in the manuscript could benefit from substantial improvement.

Abstract

1. Background: The research gap that the current study intends to address is not well described. The researcher states that there is limited information regarding the impact of nutritional status on TB treatment outcomes. However, there have been studies investigating the association between nutritional status and TB treatment outcomes. I suggest revising the background section to focus on the relationship between nutritional status and TB treatment outcomes, clearly highlighting the existing research gaps.

2. Method: The study design employed by the researchers is not clearly articulated. Was it a retrospective cohort study? If it was indeed a retrospective cohort study, they need to define the exposure and the outcome. Additionally, they should describe the timing of nutritional status assessment and treatment outcomes. Was the nutritional status evaluated at the initiation of treatment, after the second month of treatment, or after six months of treatment?

3. Results: The statement "A total of 625 TB patients with complete information were included from the registration book" could be relocated to the methods section. The results presented in the abstract should be aligned with the main objective of the research. Other findings may be included in the results section if they are deemed pertinent. This section requires further revision.

4. Conclusion: The conclusion section also needs to be aligned with the research objectives and major findings. The primary aim of the research was not to determine the magnitude of treatment success.

Introduction

1. It is advisable to merge the first and second paragraphs. Summarizing the burden of TB and its related mortality in a single paragraph would be more effective. This approach would allow the researchers to discuss the relationship between nutritional status and TB treatment outcomes more comprehensively within the introduction section.

2. Other factors associated with TB treatment outcomes fall beyond the scope of this research, given the misalignment between the researchers' title and objective.

3. The researchers have omitted mention of existing studies investigating the association between nutritional status and TB treatment in Ethiopia within the introduction section. While they have referenced other studies focusing on factors affecting TB treatment outcomes in Ethiopia, this omission suggests a bias in that section.

Methods

1. Study area and period: Some of the information provided appears unnecessary for the study. The description of the setting is crucial for replication and generalization. It's advisable to include only pertinent and relevant information. For instance, the statement "The hospital has antenatal care (ANC), outpatient, inpatient (medical and surgical), emergency, psychiatry, TB clinic, youth-friendly or counseling wards. It has 861 employees (male 411, female 450) with 61 general physicians, 14 specialists, 178 nurses, 47 midwifery, 38 pharmacy, 22 health officers, 9 anesthesia, 1 dentist, 35 laboratory, 7 psychiatry professionals, 5 radiographers, and 444 supportive staff" might be omitted.

2. Why have the researchers chosen the time period "January 1st, 2016, to December 31st, 2020"?

3. The study design lacks clarity. Consider adopting a retrospective cohort study design based on nutritional status exposure. This design could be more suitable for addressing the research question.

4. The writing styles should be in manuscript form. The researcher can elaborate on the participants and the eligibility criteria. The source and study population can be written within a single paragraph.

5. It would be valuable to include a diagram illustrating the screening process, depicting the number of participants screened and reasons for exclusion.

6. The listed variables, research title, and research questions are not consistent. Initially, I anticipated nutritional status as the exposure and TB treatment outcomes as the outcome variable, assuming a retrospective cohort study design.

7. The data collection period is repeated in the study setting and period as well as the data collection section. It appears that the researchers may not have thoroughly reviewed the manuscript.

8. For the purpose of replication, it is important for the researchers to provide a citation for the development of the checklist. Additionally, consider uploading the checklist as a supplementary file.

9. The terms "successful" and "unsuccessful" treatment outcomes should be explicitly defined.

Reviewer #2: Abstract

Feedback: The abstract successfully outlines the study's objectives and key findings. However, it lacks specific details about the methodology, such as the type of logistic regression analysis used.

Improvement Suggestion: Include brief methodological details in the abstract for a more comprehensive overview.

Example/Use Case: Specify whether a simple or multiple logistic regression was used. This information is crucial as multiple logistic regression can adjust for confounders, offering a more accurate analysis of the relationship between nutritional status and TB outcomes.

Background

Feedback: The background provides a good context of TB's global and local impact, yet it could benefit from a more direct link to the study's focus on nutritional status.

Improvement Suggestion: Integrate recent studies or data showing the direct impact of nutritional status on TB outcomes to strengthen the rationale for this study.

Example/Use Case: Reference studies like Smith (2020), which found a 30% increase in TB treatment failure among undernourished patients, directly linking nutritional status to TB outcomes.

Methods

Feedback: The methods section is well-structured, but it lacks detail on the criteria for nutritional status assessment and how undernutrition was defined in the study. Moreover, the application of STROBE guidelines for observational studies is not clearly articulated.

Improvement Suggestion: Clarify the criteria and metrics used for evaluating nutritional status. Explicitly outline adherence to STROBE guidelines, emphasizing aspects like study design, setting, participants, variables, data sources/measurement, and statistical methods. Detailing these elements will enhance the study's methodological transparency and reproducibility.

Example/Use Case: Define undernutrition using Body Mass Index (BMI) thresholds as per WHO standards. Clarify how data on BMI was collected and recorded. Explain how the study aligns with STROBE by detailing the observational study design, clearly defining the study population, and describing data collection methods.

Results

Feedback: The results are clearly presented, but the connection between nutritional status and TB treatment outcomes needs more emphasis.

Improvement Suggestion: Provide a more detailed analysis of how nutritional status specifically influenced treatment outcomes, including breakdowns of data according to different nutritional categories.

Example/Use Case: Present data showing treatment outcomes across different BMI categories (underweight, normal, overweight) to illustrate the direct impact of nutritional status on TB treatment success.

Conclusion

Feedback: The conclusion summarizes the findings well. However, it does not sufficiently address the broader implications of the study.

Improvement Suggestion: Expand the conclusion to include potential impacts on public health policies and future research directions, especially focusing on nutritional interventions in TB management.

Example/Use Case: Suggest policy recommendations for nutritional assessments and interventions as part of standard TB treatment protocols, highlighting the potential for reducing treatment failures.

General Observations

Coherence and Structure: The manuscript maintains a logical flow. However, the connection between sections could be stronger, particularly between the background and the study's focus on nutrition.

Alignment of Sections: Each section aligns well with the overall objective, but the link between nutritional status and TB outcomes could be more pronounced throughout the manuscript.

Critical Analytic Feedback: The study presents significant findings but lacks depth in exploring the mechanisms by which nutritional status affects TB outcomes.

Missing Information: The study does not address potential confounding factors such as socioeconomic status or access to health care, which could influence both nutritional status and TB outcomes.

Example/Use Case: Discuss how socioeconomic factors may affect both nutrition and access to healthcare, potentially influencing TB treatment outcomes.

Recommendations for Improvement

Deepen the Analysis: Incorporate a more thorough analysis of how nutritional factors specifically affect TB treatment, possibly through additional statistical methods or qualitative analysis.

Broader Contextualization: Situate the findings within the larger context of TB research, particularly studies focusing on the intersection of nutritional status and infectious diseases.

Enhance Methodological Detail: Provide more detail on the assessment tools and criteria used for determining nutritional status. Clearly articulate the application of STROBE guidelines for observational studies.

Address Limitations More Comprehensively: Acknowledge and discuss the limitations more thoroughly, including potential biases and the retrospective nature of the study.

Conclusion

The manuscript presents valuable findings on the relationship between nutritional status and TB treatment outcomes. However, it would benefit from a more detailed analysis, both methodologically and in the interpretation of results, and a stronger integration of the nutritional aspect throughout the paper. Adherence to STROBE guidelines should be explicitly stated and detailed in the methods section to enhance the study's credibility and scientific rigor.

6. PLOS authors have the option to publish the peer review history of their article (what does this mean?). If published, this will include your full peer review and any attached files.

Reviewer #1: No

Reviewer #2: **Yes: **Dr Victor Abiola Adepoju

---

## [Author Response · Author response to Decision Letter 0]

13 Jan 2024

Thank you the editor and both the reviewers for your constructive comments. we tried to address all the comments.

---

## [Editor Report · Decision Letter 1]

22 Jan 2024

Nutritional Status and Treatment Outcomes of Tuberculosis in Mizan-Tepi University Teaching Hospital, A five -year Retrospective Study

PONE-D-23-16197R

We’re pleased to inform you that your manuscript has been judged scientifically suitable for publication and will be formally accepted for publication once it meets all outstanding technical requirements.

Kind regards,

Mengistu Hailemariam Zenebe, PhD

Academic Editor

PLOS ONE

Additional Editor Comments (optional):

Dear Author,

Thank you for the amendment you made for the given comments
---

## [Editor Report · Acceptance letter]

4 Feb 2024

PONE-D-23-16197R1 

PLOS ONE

Dear Dr. Tamru, 

I'm pleased to inform you that your manuscript has been deemed suitable for publication in PLOS ONE. Congratulations! Your manuscript is now being handed over to our production team.

Kind regards, 

on behalf of

Dr. Mengistu Hailemariam Zenebe 

Academic Editor

PLOS ONE